# Successful Treatment of Catastrophic Antiphospholipid Syndrome Using Rituximab: Case Report and Review of the Literature

**DOI:** 10.3390/medicina57090912

**Published:** 2021-08-31

**Authors:** Cristina Stanescu, Andreea Gabriella Andronesi, Ciprian Jurcut, Mihaela Gherghiceanu, Alexandra Vornicu, Florentina Andreea Burcea, Toader Danut Andronesi, Gabriela Elena Lupusoru, Luminita Iliuta, Bogdan Marian Sorohan, Bogdan Obrisca, Gener Ismail

**Affiliations:** 1Nephrology Department, Fundeni Clinical Institute, 022328 Bucharest, Romania; cristina.cristache77@yahoo.com (C.S.); vornicu.alexandra@yahoo.com (A.V.); andreea.burcea11@gmail.com (F.A.B.); gabriela.topor@yahoo.com (G.E.L.); gener732000@yahoo.com (G.I.); 2Nephrology Department, “Carol Davila” University of Medicine and Pharmacy, 050471 Bucharest, Romania; bogdan.sorohan@yahoo.com (B.M.S.); obriscabogdan@yahoo.com (B.O.); 3Internal Medicine Department, “Carol Davila” Military Emergency Hospital, 010225 Bucharest, Romania; cjurcut@gmail.com; 4“Victor Babes” National Institute for Research and Development in Pathology and Biomedical Sciences, 050097 Bucharest, Romania; mgherghiceanu@yahoo.com; 5Department of General Surgery and Liver Transplantation, Fundeni Clinical Institute, 022328 Bucharest, Romania; dan_andronesi@yahoo.com; 6Department of Biostatistics, Marketing and Medical Technology, “Carol Davila” University of Medicine and Pharmacy, 050471 Bucharest, Romania; luminitailiuta@yahoo.com

**Keywords:** lupus, catastrophic antiphospholipid syndrome, rituximab, thrombotic microangiopathy, case report

## Abstract

Background: Kidney involvement is a frequent complication of systemic lupus erythematosus (SLE) and kidney biopsy is essential in differentiating lupus nephritis (LN) from thrombotic microangiopathy (TMA) secondary to antiphospholipid autoantibodies (aPL). Association between antiphospholipid syndrome (APS) and acquired hemophilia due to inhibitors was very rarely described in SLE patients. Case presentation: We present the case of a 61-year-old male diagnosed with SLE who acquired deficiency of clotting factor VIII due to circulating inhibitors, admitted for acute kidney injury (AKI), microangiopathic hemolytic anemia, thrombocytopenia, and diplopia. Kidney biopsy showed TMA due to APS, but no signs of LN. Head computed tomography identified low dense areas in the white matter, suggesting small blood vessels’ involvement. A diagnosis of probable catastrophic antiphospholipid syndrome (CAPS) was established and treatment with low molecular weight heparin, intravenous methylprednisolone, plasmapheresis, and rituximab was initiated, followed by resolution of AKI, diplopia, and TMA with complete depletion of CD19+B-lymphocytes (CD19+B-Ly) after one month. We further review the current knowledge regarding pathogenesis and management of CAPS in SLE patients. Conclusions: Targeted therapy was possible after kidney biopsy, improving renal and general prognosis. CD19+B-Ly repopulation preceded biological relapse, so monitoring of CD19+B-Ly may serve as a tool to predict relapses and guide rituximab therapy.

## 1. Introduction

Antiphospholipid syndrome (APS) was first described by Graham Hughes in 1983–1985. The incidence is about 20% in patients under 50 years who suffered a stroke, 12–30% in systemic lupus erythematosus (SLE), and 10–15% in women with repeated miscarriages [1]. The clinical spectrum of antiphospholipid (aPL) antibodies starts from the simple positivity, with no clinical events or positive aPL with non-diagnostic criteria (such as livedo reticularis, thrombocytopenia, microangiopathic hemolytic anemia, valve abnormalities, aPL associated nephropathy, and chorea) to APS and catastrophic antiphospholipid syndrome (CAPS).

Thrombotic microangiopathy (TMA) implies a pathological process secondary to microvascular occlusion due to platelets’ aggregates, causing thrombocytopenia and microangiopathic hemolytic anemia. There are hereditary or acquired disorders, such as CAPS. CAPS is a life-threatening systemic disease that complicates around 1% of APS. Most frequently, kidneys are involved (73%), followed by lungs (59%), central nervous system (56%), and heart (50%), but also the intestines, spleen, pancreas, adrenal glands, and bone marrow may be targeted [2]. Around 40% of all CAPS cases have an associated autoimmune condition, with SLE being the most frequent one; although a rare condition, CAPS may have a significant impact upon patients, with a fatal outcome in up to almost 40% of all patients [3]. SLE-associated CAPS usually has a more severe evolution, with frequent brain and heart involvement and mortality in half of the patients [3].

We present a case of CAPS secondary to SLE in an elderly male patient in whom a favorable outcome was obtained through a multidisciplinary approach. The written informed consent of the patient was obtained in order to publish this case.

## 2. Case Presentation

A 61-year-old Caucasian male patient was admitted to our nephrology department on March 2016 for acute kidney injury (serum creatinine (SCr) 1.6 mg/dL, compared with 0.9 mg/dL one month before). He was diagnosed in 2009 with benign polyclonal gammopathy and in 2012 with persistent double positivity for aPL without clinical manifestations (IgM anticardiolipin antibodies (aCL) in a titer of 43.6 MPL/mL, and positive lupus anticoagulant—confirmed by mixing studies and demonstration of phospholipid dependence), with overlap syndrome (primary biliary cirrhosis-autoimmune hepatitis), with acquired factor VIII deficiency due to inhibitory antibodies, and with severe aortic stenosis. Surgery for valvular heart disease was declined by the surgical team because of coagulopathy. He developed pancytopenia associated with a positive Coombs test, low C3 and C4 complement fractions, and positivity for cryoglobulins in 2014, with rapid resolution of blood abnormalities after a short course of steroids. Aortic valve replacement with a biologic valve was performed in 2015 with correction of blood level of factor VIII using steroids prior to surgery. His family medical history is relevant for a diagnosis of rheumatoid arthritis in his mother.

Prior to the admittance in our clinic, he was evaluated in an internal medicine department for low back pain with inguinal extension and self-limited gross hematuria. Raised SCr was noticed for the first time together with intense positivity for double stranded DNA (dsDNA) at a titer of 225 U/L. The patient was diagnosed with SLE, hydroxychloroquine 200 mg BID was initiated, and he was referred to our clinic.

At admittance, the clinical exam was unremarkable. Progressing renal dysfunction (SCr 1.9 mg/dL), raised uric acid (10.2 mg/dL), and inflammatory syndrome (erythrocytes sedimentation ratio 64 mm/h, C-reactive protein 42.5 mg/L, with normal serum fibrinogen level) were identified. Urinalysis showed numerous urate crystals and no dysmorphic red blood cells or cellular casts; proteinuria was 324 mg/day. The abdominal ultrasound showed grade I left hydronephrosis and the Doppler exam revealed normal kidneys’ vascularization. The non-enhanced abdominal computed tomography (CT) scan identified multiple bilateral kidney stones, but no ureteral calculi. Allopurinol 150 mg/day was initiated. A kidney biopsy was considered, but we decided to postpone it because of hydronephrosis.

The patient came back after one month with short periods (2–3 min) of diplopia during the past week and worsening of renal dysfunction (SCr 2.3 mg/dL) and of proteinuria (800 mg/day), with bland urine sediment, newly developed thrombocytopenia (79,000/mL), and mild normocytic normochromic anemia (hemoglobin 11.8 g/dL). A head CT scan showed small low dense areas in the frontal and parietal white matter, suggesting a possible involvement of small blood vessels. Activity of the coagulation factor VIII was 20%, so a short course of intravenous methylprednisolone 250 mg/day for 6 days (continued with oral methylprednisolone 64 mg/day) was started followed by a percutaneous biopsy of the left kidney. With steroids, kidney function improved (SCr 1.27 mg/dL at discharge). 

Although kidney dysfunction was a recent diagnosis, light microscopy showed global glomerular sclerosis of three out of the five glomeruli analyzed, together with moderate interstitial fibrosis and rare areas with infiltration of lymphocytes (Figure 1).

An important finding was the presence of an arteriole with luminal shrinkage and endothelial edema, which are lesions highly suggestive of endotheliosis (Figure 2), together with the presence of multiple thrombi in the lumen of arterioles, the hallmark of renal TMA (Figure 3).

Electron microscopy confirmed intravascular thrombosis, with cellular fragments and acellular, fibrinous material inside the arteriolar lumens (Figure 4). The immunofluorescence was negative (Figure 5).

The final pathological diagnosis was aPL-associated nephropathy.

The patient came back after three weeks with further degradation of kidney function (SCr 3.1 mg/dL) and signs of microangiopathic hemolytic anemia (peripheral blood smear showing erythrocytes’ fragmentation). Thrombotic thrombocytopenic purpura (TTP) was ruled out (normal Willebrand factor-cleaving protease—ADAMTS13 activity in the plasma). We performed a genetic testing for major histocompatibility class genes, which showed a high risk profile for several autoimmune diseases (Table 1).

A diagnosis of probable CAPS diagnosis was established, as all four diagnostic criteria were met, but with only two organs affected (aPL-associated nephropathy and cerebral microangiopathy). Because we confirmed the presence of small vessel occlusion through kidney biopsy, we did not perform other pathological examination. The patient received anticoagulation with subcutaneous low molecular weight heparin-LMWH (dalteparin 5000 IU daily), intravenous methylprednisolone 250 mg for four days, followed by oral methylprednisolone 0.5 mg/kg/day. Five plasma exchange (PLEX) sessions every other day were performed, followed by two Rituximab infusions of 500 mg seven days apart. At discharge, SCr was 1.3 mg/dL, with no hemolysis. Treatment was continued with oral methylprednisolone with gradual tapering over the next six months, as well as hydroxychloroquine, dalteparin (we choose not to switch to oral antivitamin K (AVK) owing to acquired factor VIII deficiency), together with anti-osteoporosis treatment with Denosumab and alpha vitamin D. Complete CD19+ B cell depletion was obtained one month after Rituximab. The aCL titer progressively decreased, with negative results after five months. Kidney function remained normal during the follow-up of 40 months. Eighteen months after rituximab administration, we observed the repopulation of CD19+ B cells, followed by positivity of aCL two months apart, but without signs of hemolysis. Then, 500 mg of Rituximab was administered with complete CD19+ B cell depletion and aCL negativity. Regarding LA, it remained positive during the whole follow-up. The treatment was well tolerated, without any side effects.

## 3. Discussion

### 3.1. Catastrophic Antiphospholipid Syndrome

CAPS is a rare variant of APS that occurs in less than 1% of the APS patients, with up to 50% mortality [3,5,6]. It is a form of TMA and implies a pathological process secondary to microvascular occlusion owing to platelets’ aggregates, causing thrombocytopenia and microangiopathic hemolytic anemia. Most CAPS cases occur in young females, with elderly males rarely being affected [3], as happened in our patient.

Multiple pathogenic mechanisms are involved in CAPS, such as cellular activation, inhibition of anticoagulants and of fibrinolysis, and complement activation (including complement regulatory gene mutations) [6,7]. A number of precipitating factors have been identified in around two-thirds of all CAPS episodes such as infections, malignancies, surgical interventions, and disease activity of SLE [3,8]. All these factors may lead to the occurrence of thrombosis and of systemic inflammatory response syndrome (SIRS). No precipitating factor was identified in our patient; regarding his previous surgery for heart valve replacement, this happened 10 months before he developed the first signs of CAPS, so no connection can be established between these two conditions. Genetic factors may also play a role in the development of CAPS. According to a systematic review, 16 genes (including toll-like receptor 2 and 4, platelet glycoprotein Ib alpha chain, coagulation factor II thrombin receptor, tissue factor pathway inhibitor, coagulation factor III, vascular endothelial growth factor A, and TNF) were identified in patients with primary APS in comparison with controls. Thirty-two organs (including the kidneys, the liver, the lungs, the colon, the cerebral cortex, and the gall bladder) express these genes, predisposing them to a higher risk of developing thrombosis [9]. Population studies on aPL in diseases other than primary APS, such as SLE, have shown an association with certain genetic patterns (Table 2) [10,11]. MHC class II testing in our patient revealed a genotype highly susceptible to autoimmune diseases (Table 1).

### 3.2. APS-Associated Nephropathy

At presentation, only 18% of patients have kidney involvement, but, eventually, up to 80% of them will develop kidney disease [12,13]. At least one of the following lesions needs to be identified at kidney biopsy for the diagnosis of APS nephropathy: acute lesion of TMA, arterial and arteriolar recanalizing thrombi, interlobular fibrous intimal hyperplasia, fibrous arterial occlusion, or focal cortical atrophy [13]. In a cohort from Thailand of 150 patients with LN, 34% also had APS-associated nephropathy. These lesions were correlated with the disease activity and chronicity indices, hypertension, severe proteinuria, renal failure, class III and IV histology, and end-stage renal disease (ESRD) [14]. The activation of the complement cascade is an important mechanism [15] and the absence of complement regulatory proteins on glomerular cells is associated with TMA [16].

ESRD is not common in APS. In a prospective study that analyzed 39 APS patients over a period of 10 years, only one patient developed ESRD [17], while another study, in which 20 consecutive APS patients were evaluated, of whom five presented histological lesions of acute or chronic TMA, demonstrated that two of these five patients progressed to ESRD [18]. Nevertheless, patients with ESRD, no matter the underlying cause of kidney disease, have a higher incidence of aPL positivity when compared with the general population [19]. aPLs increase the risk of thrombosis in those who undergo kidney transplantation [20]. More importantly, anticoagulation therapy does not seem to completely prevent graft loss [21].

Nowadays, a consensus on the treatment of APS nephropathy does not exist. These patients should receive the standard anticoagulation treatment of APS, although, if APS nephropathy is present in the absence of definite APS criteria, no specific regimens are established. Those who also present LN should receive hydroxychloroquine and immunosuppressive treatment [22]. The novel oral anticoagulants do not have studies in APS nephropathy.

### 3.3. CAPS Treatment

Precipitating factors, such as infections and malignancies, must be addressed [23]. The mortality of CAPS was over 50% before the 2000s, as most of the patients received only anticoagulation, while these days, mortality decreased to around one-third as a result of the shift in the treatment paradigm, which now consists of anticoagulation, steroid therapy, and PLEX or intravenous immunoglobulins (IVIg) [3,24,25]. In the analysis of Bucciarelli, the higher rate of use of combined triple therapy was an independent predictive factor for the decrease in mortality observed after the year 2000 [25]. The mainstay in CAPS treatment is anticoagulation with heparin [6], as it has the most important effect on patients’ vital prognosis [3,24]. Moreover, it seems that the anti-inflammatory effect of heparin is of great importance in CAPS. AVK, initiated after heparin, is the elected anticoagulant, and it should be continued long term. Even though more information about the use of direct anticoagulants (DOACs) is emerging, there is not enough evidence of their effectiveness and safety in patients with CAPS [26,27]. In our patient, we could not initiate the AVK treatment given the intrinsic coagulopathy, so he received long-term treatment with LMWH.

Regarding steroid therapy, intravenous methylprednisolone (usually 1000 mg for three consecutive days) followed by oral steroids with gradual tapper is used [2]. Most experts recommend using steroids in CAPS for their anti-inflammatory effects.

PLEX is an established therapeutic procedure in CAPS, but there is not enough information regarding the choice of replacement fluid, timing, and number of sessions in patients with CAPS, and there are no clinical trials on the effectiveness of PLEX in CAPS. Current guidelines include the use of therapeutic PLEX, especially in patients with microangiopathic features (schistocytes) or CAPS-associated kidney involvement, and it was observed that those who underwent this procedure had a lower mortality rate [24,28]. Five PLEX sessions every other day were performed in our patient and rituximab was administered only after last PLEX session, as this drug is susceptible for being removed from the plasma during the apheresis procedures.

Solely using IVIg (0.4 g/kg/day for 4–5 days) does not seem to lower mortality, but is beneficial in patients with immune-mediated thrombocytopenia, and they must be administered only after the completion of PLEX [28].

There are alternative therapies in CAPS, including cyclophosphamide and hydroxychloroquine in refractory CAPS with active SLE, the latter lowering the activation of thrombocytes, diminishing the aPL-betaglycoprotein I (aPL-βGPI) complex attachment to membrane, enabling the expression of annexin A5, inhibiting toll-like receptors, and lowering the thrombotic and cardiovascular risk in SLE [29]. Defibrotide modulates the activity of TNF-α, endothelin, thrombin, and interleukin-2, and it also has a beneficial effect on endothelial dysfunction, demonstrating antithrombotic, anti-ischemic, and anti-inflammatory effects, with no anticoagulant activity [29]. Unfortunately, data regarding the use of defibrotide in CAPS cases are only limited to very few patients treated. A case report illustrated a CAPS patient with only limited response to heparin, aspirin, and dipyridamole, who had complete remission after defibrotide [30]. Another middle-aged female patient was treated with anticoagulation and defibrotide, but did not respond to this approach and died [31].

Rituximab induces B cell depletion, which, in turn, creates a biphasic mechanism of immunosuppression [32]. Firstly, it stops the production of pathogenic IgG autoantibodies and promotes the appearance of the regulatory B cell compartment, which synthetizes interleukin-10, which acts as a negative sensor of the immune response in autoimmunity [33]. The titer of IgG aCL significantly decreases after rituximab and cyclophosphamide [34]. An uncontrolled, non-randomized prospective pilot study evaluated the efficacy and safety of rituximab in the treatment of non-criteria manifestations in patients with classic APS [34]. Rituximab may have effects in controling some of these criteria, like thrombocytopenia or skin ulcers. It must be mentioned that this study failed to show substantial change in aPL profile after a 12-month follow-up. Some of the features of CAPS may be attributed to SIRS, which is associated with high levels of acute phase reactants and cytokines (e.g., TNF-α, IL-1, IL-2, IL-6). By reducing the number of circulating B cells, rituximab may also lower the levels of these cytokines [30]. The 20 patients from the CAPS registry who received Rituximab had a higher chance of recovery (75%), in comparison with Bucciarelli’s cohort (65%) [25], suggesting that Rituximab may improve the patients’ outcome [35]. In 8 out of the 20 patients from the CAPS registry, Rituximab was used as a first-line treatment together with combined therapy (like in our patient), either because of the severity of initial manifestations or owing to associated lymphoma; in the other 12 patients, Rituximab was used as a second-line treatment, because of refractory first CAPS episode or recurrent CAPS [36]. Owing to the rarity of the disease and because there have been only a few patients treated with Rituximab until now, there is no consensus regarding the best dosing regimen—the most used ones were either four weekly doses of 375 mg/m^2^ or two infusions of 500–1000 mg at 7 or 14 days apart [3,12,37]. We chose the second dosing regimen in our patient and Rituximab was administered as an add-on therapy to combined triple therapy because of the good safety profile and accompanying thrombocytopenia and proved APS nephropathy. We considered our patient to have a potentially severe CAPS owing to brain and kidney involvement and, for this reason, we wanted a prolonged lymphocytes B suppression. Monitoring B cell depletion after Rituximab may be used as a surrogate marker for relapse prediction and individualized treatment in several autoimmune diseases, including rheumatoid arthritis [38], multiple sclerosis [39], and certain kidney diseases (including steroid-dependent nephrotic syndrome, membranous nephropathy, and antibody-mediated kidney graft rejection) [40]. Complete CD19+ depletion was obtained one month after Rituximab in our patient, but 18 months later, the repopulation occurred, followed by aCL positivity after two more months. Another 500 mg of Rituximab was administered with complete CD19+ B cell depletion, disappearance of aCL, and no recurrence of CAPS. Data regarding the response of aPL after Rituximab infusions are even more scarce. In the CAPS registry, this information was available for only eight patients, with four patients obtaining aPL negativity and four patients maintaining positivity (but, even in these four patients, the status of all the three aPL types was reported in only one case) [36]. In our patient, aCL became negative five months after Rituximab infusions, with positivity 2 months after CD19+ B cells’ repopulation and negativity again after Rituximab repeated administration, while LA remained positive all the time. Of course, it is impossible to analyze the isolated effect of B-cell depleting therapy upon CAPS evolution and aPL profile, as all of the patients from the CAPS registry, as well as our case, received a combined therapy.

Eculizumab, a humanized monoclonal antibody against complement protein C5, was proposed as an alternative treatment in CAPS; however, its role is still debatable [41]. One case report showed a positive result after Eculizumab use for the prevention of CAPS recurrence after kidney transplantation [42], while other papers reported successful treatment with Eculizumab in patients with refractory CAPS [43,44,45]. Two open-label studies were conducted to analyze Eculizumab for APS/CAPS therapy. One of the trials evaluated if blocking the complement in patients with history of CAPS who undergo renal transplantation would lead to better transplant outcomes [46]. The other one evaluated the safety and tolerance of an intravenous C5 inhibitor in patients with persistently positive aPL and at least one non-criteria manifestations of APS [47]. However, both trials were stopped prematurely, the first one because only one patient could be enrolled, and the second one because of slow patient enrollment. Eculizumab may play a role in the treatment of CAPS patients who associate complement-related regulatory gene mutations, but this issue needs to be addressed in future studies [48].

### 3.4. Acquired Hemophilia A

It is a very rare condition (1.5 per million per year) caused by the presence of inhibitor antibodies [49]. Approximately half of the cases appear in healthy individuals, but the other half develops in association with the postpartum period, autoimmune diseases, cancers, or use of certain drugs. Our patient had a very rare association, between aPL and acquired FVIII deficiency, with only a few cases reported in the literature. This association raised serious problems regarding increased risk for bleeding during surgical maneuvers (both heart valve replacement and kidney biopsy) and optimal anticoagulation for CAPS. Occurrence of both APS and acquired hemophilia is critical to identify, as the manifestations can range from massive hemorrhage to thrombosis.

## 4. Conclusions

Besides the complexity and severity of the presented case, the importance of renal biopsy and the interdisciplinary approach of the patient’s therapy must be noted. Renal biopsy was of central importance in establishing the correct histological diagnosis, allowing the proper treatment and thus improving the renal and general prognosis in a rare, but very severe disease, such as CAPS. Moreover, the extremely rare association of APS and acquired hemophilia imposed a careful consideration of the therapy for the prevention of thrombosis, as well as the possible bleeding complications. Even though the patient always had a good clinical status, the diagnosis of probable CAPS needed immediate attention and aggressive treatment, as this disease has a high rate of mortality even with the maximal therapy available nowadays. Rituximab may be used as an add-on therapy in patients with severe, refractory, or recurrent CAPS. Monitoring CD19+ B-cells may be a useful tool in tailoring B-cell depleting treatment, and thus prevents potential CAPS relapses; this approach needs to be investigated in further prospective studies.

## Figures and Tables

**Figure 1 medicina-57-00912-f001:**
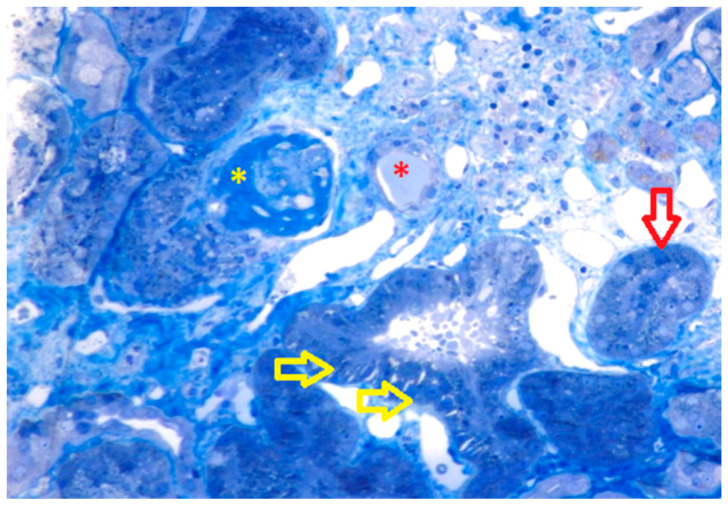
Light microscopy. Global glomerulosclerosis (yellow asterisk); tubular epithelium with cellular edema and vacuolization (yellow arrows); a tubular lumen with amorphous, acellular material inside (red asterisk); and a tubule with edematous epithelium and complete luminal obstruction by fibrosis and cellular fragments. In the upper right corner, minimal inflammation can be noticed, highlighted by the presence of dark blue neutrophils. In the lower left corner, the bright blue pattern of the interstitium suggests fibrosis (toluidine blue, 40× magnification).

**Figure 2 medicina-57-00912-f002:**
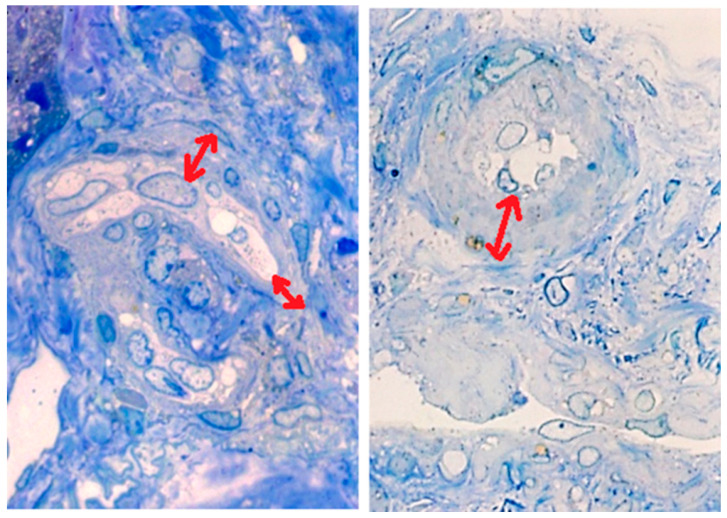
Light microscopy. Lesions suggesting endotheliosis luminal shrinkage and endothelial hyperplasia (red double arrows) (toluidine blue, 40× magnification).

**Figure 3 medicina-57-00912-f003:**
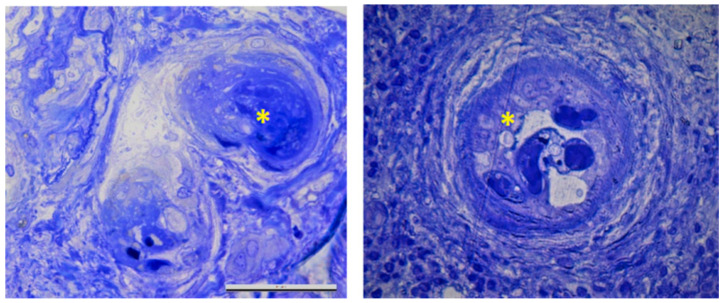
Light microscopy. Thrombus inside an arteriole (yellow asterisk) sustaining the diagnosis of renal thrmboticmicroangiopathy (blue toluidine, 40× magnification).

**Figure 4 medicina-57-00912-f004:**
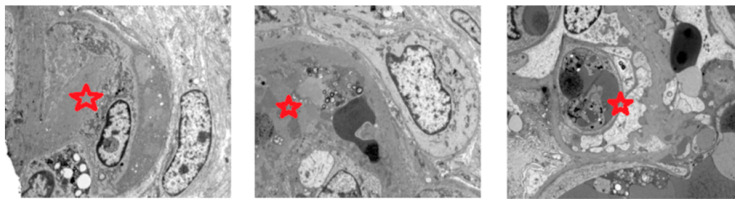
Electron microscopy. Fibro-cellular thrombi (red stars) inside the arterioles.

**Figure 5 medicina-57-00912-f005:**
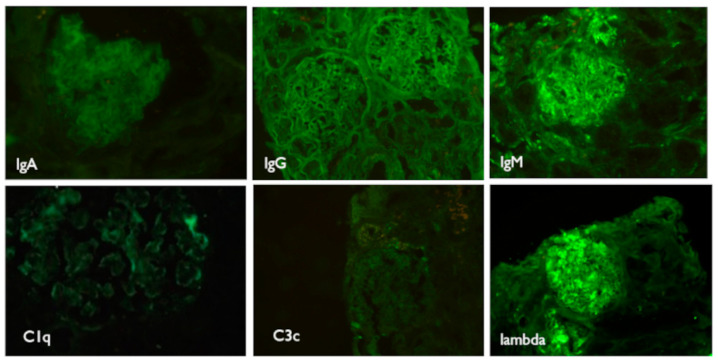
Immunofluorescence. Weak IgA positivity in GBM and mesangial areas. Negative IgG. Weak positivity fine granular IgM. Negative C1q and C3. Moderate positivity for lambda chains with granular aspect. IgA—immunoglobulin A; GBM—glomerular basement membrane; IgG—immunoglobulin G; IgM—immunoglobulin M.

**Table 1 medicina-57-00912-t001:** The genetic testing of our patient revealed multiple high-risk gene patterns for autoimmune diseases [4].

MHC Class II	
DQB1: *02, *05	DQB1*02 risk factor for SLE, type 1 diabetes mellitus
DRB1: *03, *14	DRB1*03 risk factor for SLE; multiple sclerosis Hashimoto/Graves’ disease, type 1 diabetes mellitus
	DRB1*14 risk factor for rheumatoid arthritis.

MHC—major histocompatibility complex; SLE—systemic lupus erythematosus.

**Table 2 medicina-57-00912-t002:** Genotypes linked to antiphospholipid syndrome predisposition [10,11].

DQB1*0604/5/6/7/9-DQA1*0102-DRB1*132
MHC-DQw7 (allele DQB1*0301)
MHC-DR5
MHCDRB1*03
MHC-DMA*0102
TNFA-238*A-DGB1*0303-DRB1*0701

MHC—major histocompatibility; TNFA-238*A—tumor necrosis factor alpha-238 A.

## Data Availability

The data presented in this study are available on request from the corresponding author. The data are not publicly available due to privacy issues.

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
