# Peer review of "Successful Treatment of Catastrophic Antiphospholipid Syndrome Using Rituximab: Case Report and Review of the Literature"

_medicina, 2021, doi:10.3390/medicina57090912_

Round 1

Reviewer 1 Report

I want to congratulate Doctor Stanescu and team for this case report. Data are scarce on that topic and it adds precious results. It is well writing.

However some important points need to be clarify or modify.

Page 2 when describing CAPS, please add the data from the CAPS registry since it is the most important article about this topic. I am not sure that it is pertinent to describe APS in women with repeated miscarriages since it is not this cluster who experiments CAPS, but more women with fetal death and HELLP syndrome.

Concerning case presentation, had the APS biology been controlled at 12 weeks? what about the Rösner index ?

Please describe if there was previous event which could be precipitating factors like when was the surgery in 2015 (maybe at the end which can be close to the beginning of the symptoms).

The patient had biological lupus (Low C3 and anti DNA positivity) which led to the prescription of HCQ but did not fulfilled the international criteria for SLE, please modify “ patient was diagnosed with SLE”.

About the first CT scan for back pain, did it show infarctus ? maybe specify it.

Page 5 what about schistocytes and platelets ?

Did the patient undergo a cutaneous examination to search ungual hemorrhage for example?

What about LA after rituximab pulse ?

For recommendation of treatment, the authors have to add Rodriguez-pinto 2016 “Catastrophic antiphospholipid syndrome: The current management approach » to the discussion. Moreover, the recommendations are IVIg or plasma exchange and often Plasma exchange are preferred when there is a renal involvement.

I think that it is not an update to speak about dipyridamole from an article writing in 2002.

Page 8 it is not alternative therapies in CAPS even in refractory CAPS cyclophosphamide and HCQ. It has shown only benefit if associated with lupus nephritis.

The treatment with rituximab has to be more discussed with more recent literature.

The use of eculizumab is very discussed and there were some previous studies that have to be quote.

Finally, the use of rituximab had to be more discussed and is not to date recommended in first line treatment. The patient may also have improved with the tritherapy (anticoagulant, plasma exchange, corticosteroids).

Author Response

We thank the Reviewer 1 for the appreciation regarding our manuscript, for the time allocated for the review and for all the suggestions which will clearly help to improve our work. We further present a point by point answer to all these suggestions hoping that our changes made to the manuscript and explanations provided below will answer to all the raised concerns regarding the manuscript.

Point 1. Page 2 when describing CAPS, please add the data from the CAPS registry since it is the most important article about this topic. I am not sure that it is pertinent to describe APS in women with repeated miscarriages since it is not this cluster who experiments CAPS, but more women with fetal death and HELLP syndrome.

Please see lines 52-59; 63-67; 181-182

Point 2. Concerning case presentation, had the APS biology been controlled at 12 weeks? what about the Rösner index?

Please see lines 75-78

Point 3. Please describe if there was previous event which could be precipitating factors like when was the surgery in 2015 (maybe at the end which can be close to the beginning of the symptoms)

Please see lines 189-191

Point 4. The patient had biological lupus (Low C3 and anti DNA positivity) which led to the prescription of HCQ but did not fulfilled the international criteria for SLE, please modify “ patient was diagnosed with SLE”.

Patient fulfilled the SLICC criteria for SLE diagnosis with 7 criteria (raised ANA, low C3, aPL positivity and anti-DNAds positivity; please see line 81 for autoimmune hemolytic anemia, leucopenia, and thrombocytopenia;). We omitted in our manuscript to provide the result for Coombs test, we corrected this now (please see line 81).

Point 5. About the first CT scan for back pain, did it show infarctus? maybe specify it.

The CT scan was a non-enhanced one, as it is mentioned on line 98, so it did not offer data regarding kidney vascularization. However, we included now the results of Doppler ultrasound- lines 97-98. The whole picture (colicky pain, new onset left hydronephrosis, left kidney stones noticed at CT scan, abundant uric acid crystals in the urine) was highly suggestive of a kidney stone passage, so we decided not to proceed with further evaluations.

Point 6. Page 5 what about schistocytes and platelets?

Regarding platelets, newly developed thrombocytopenia was mentioned on line 108 and presence of schistocytes is now further explained on lines 147-148.

Point 7. Did the patient undergo a cutaneous examination to search ungual hemorrhage for example?

Please see lines 158-160

 Point 8. What about LA after rituximab pulse?

Please see lines 174, 377-378

Point 9. For recommendation of treatment, the authors have to add Rodriguez-pinto 2016 “Catastrophic antiphospholipid syndrome: The current management approach » to the discussion. Moreover, the recommendations are IVIg or plasma exchange and often Plasma exchange are preferred when there is a renal involvement.

Please see lines 245, 248-249, 303-305 and reference 24

Point 10. I think that it is not an update to speak about dipyridamole from an article writing in 2002.

Regarding reference 30 (and now also 31), it was introduced because the discussion is about defibrotide, not dipyridamole. We totally agree that the references are old, but as we mentioned in the manuscript (please see lines 318-319), there are only very few cases reported who were treated with this drug. Moreover, defibrotide is mentioned in all the articles regarding possible treatments addressing APS and CAPS refractory cases, including some very recent reviews (like Rodziewicz M, 2020 “An update on the management of antiphospholipid syndrome”) and this is the reason for which we chose to discuss this possible treatment. If you still consider inappropriate to cite this papers, we will completely remove the whole discussion about defibrotide.

Point 11. Page 8 it is not alternative therapies in CAPS even in refractory CAPS cyclophosphamide and HCQ. It has shown only benefit if associated with lupus nephritis.

We totally agree with this affirmation, for this reason we mentioned that “cyclophosphamide and hydroxychloroquine should be used in refractory CAPS with active SLE” (please see line 312) and “Those who also present LN should receive hydroxychloroquine and immunosuppressive treatment” (please see line 237).

Point 12. The treatment with rituximab has to be more discussed with more recent literature.

Please see lines 329-333, 339-347, 351-368, 371-378 and newly added references 34, 37-41

Point 13. The use of eculizumab is very discussed and there were some previous studies that have to be quote.

Please see lines 382-385, 390-394 and newly added references 43-46

Point 14. Finally, the use of rituximab had to be more discussed and is not to date recommended in first line treatment. The patient may also have improved with the tritherapy (anticoagulant, plasma exchange, corticosteroids).

Please see lines 347-351, 374-380

Thank you very much.

Corresponding author on behalf of all authors

Reviewer 2 Report

  The authors show a case report of a patient with CAPS who enters in remission after two doses of rituximab.  Rituximab is currently used in CAPS, but doses are controversial (DOI: 10.1186/s12882-018-0928-z) . This aspect might be included in the text.

Despite the role of complement un APL and/or CAPS (doi: 10.1182/blood.2019003863, DOI: 10.1007/s11926-021-00984-1 ) clinical and therapeutic implications are still debatable. The two eculizumab trials the author report as ongoing are no longer active: NCT01029587 (Ref 33) is not a great trial as it was completed last year accounting for just 1 patient. On the other hand,  NCT02128269 was terminated after recruiting 9 patients due to slow rate of recruitment. 

The manuscript is well written and easily readable. I think thar after minor amendments it might be published. 

Author Response

We thank the Reviewer 2 for the kind appreciation regarding our manuscript and for the possibility of being published after properly addressing the raised issues, for the time allocated for the review and for all the suggestions which will help to improve our paper. We further present a point by point answer hoping that our changes made to the manuscript will answer to all the raised concerns.

Point 1. The authors show a case report of a patient with CAPS who enters in remission after two doses of rituximab.  Rituximab is currently used in CAPS, but doses are controversial (DOI: 10.1186/s12882-018-0928-z). This aspect might be included in the text.

Please see lines 343-347 and newly added reference 37

Point 2. Despite the role of complement un APL and/or CAPS (doi: 10.1182/blood.2019003863, DOI: 10.1007/s11926-021-00984-1) clinical and therapeutic implications are still debatable. The two eculizumab trials the author report as ongoing are no longer active: NCT01029587 (Ref 33) is not a great trial as it was completed last year accounting for just 1 patient. On the other hand, NCT02128269 was terminated after recruiting 9 patients due to slow rate of recruitment. 

Please see lines 382-386, 390-394 and newly added references 43-46.

Thank you very much.

Corresponding author on behalf of all authors

Round 2

Reviewer 1 Report

I want to congratulate the authors for this remarkable work on this review. All the listed points were answered and the manuscript is now clearer.

I would like to remove the defibrotide paragraph since I find it still inappropriate and I would be grateful if authors can add the reference of the last EULAR recommendations for APS (EULAR recommendations for the management of antiphospholipid syndrome in adults by Maria G Tektonidou et al, 2009) in the manuscript since they listed rituximab and eculizumab  as treatment in refractory CAPS ("in patients with refractory CAPS, B cell depletion (eg, rituximab) or complement inhibition (eg, eculizumab) therapies may be considered).
